# An assessment of non-communicable disease mortality among adults in Eastern Uganda, 2010–2016

**Davis Natukwatsa**[1,2]*, **Adaeze C. Wosu**[3], **Donald Bruce Ndyomugyenyi**[1,2], **Musa Waibi**[1,2], **Dan Kajungu**[1,2]

**1** Makerere University Center for Health and Population Research, Kampala, Uganda, **2** Iganga-Mayuge Health and Demographic Surveillance Site, Iganga, Uganda, **3** Department of Epidemiology, Johns Hopkins Bloomberg School of Public Health, Baltimore, Maryland, United States of America

* natukwatsa.davis@gmail.com

**Data Availability Statement:** Data is confidential and cannot be shared publicly. Anonymized data can only be shared upon request, as per the data sharing agreement from Makerere University

## Abstract

### Background

There is a dearth of studies assessing non-communicable disease (NCD) mortality within population-based settings in Uganda. We assessed mortality due to major NCDs among persons ≥ 30 years in Eastern Uganda from 2010 to 2016.

### Methods

The study was carried out at the Iganga-Mayuge health and demographic surveillance site in the Iganga and Mayuge districts of Eastern Uganda. Information on cause of death was obtained through verbal autopsies using a structured questionnaire to conduct face-face interviews with carers or close relatives of the deceased. Physicians assigned likely cause of death using ICD-10 codes. Age-adjusted mortality rates were calculated using direct method, with the average population across the seven years of the study (2010 to 2016) as the standard. Age categories of 30–40, 41–50, 51–60, 61–70, and ≥ 71 years were used for standardization.

### Results

A total of 1,210 deaths among persons ≥ 30 years old were reported from 2010 to 2016 (50.7% among women). Approximately 53% of all deaths were due to non-communicable diseases, 31.8% due to communicable diseases, 8.2% due to injuries, and 7% due to maternal-related deaths or undetermined causes. Cardiovascular diseases accounted for the largest proportion of NCD deaths in each year, and women had substantially higher cardiovascular disease mortality rates compared to men. Conversely, women had lower diabetes mortality rates than men for five of the seven years examined.

### Conclusions

Non-communicable diseases are major causes of death among adults in Iganga and Mayuge; and cardiovascular diseases and diabetes are leading causes of NCD deaths.

Center for Health and Population Research (MUCHAP). The Iganga Mayuge Health and Demographic Surveillance Site (IMHDSS) is bound to a data sharing agreement with the larger INDEPTH network health and demographic surveillance sites. There is a formal data sharing process guided by data sharing standard operating procedures. Data can be requested formally from the IMHDSS leader (info@muchap.mak.ac.ug) who is the point of contact for all data requests. There are more details on the website (www.muchap. mak.ac.ug)."

**Funding:** The author(s) received no specific funding for this work.

**Competing interests:** The authors declare no competing interests.

Efforts are needed to tackle NCD risk factors and provide NCD care to reduce associated burden and premature mortality.

## Introduction

Non-communicable diseases (NCDs) are serious global health challenges, particularly in low and middle income countries (LMICs) where rates of these conditions are rising due to significant and rapid changes in social and behavioural risk factors, and health systems are often overstretched [1]. Global efforts to curb the rise of NCDs include the WHO global action plan for prevention and control of NCDs which was established in 2013 [2], and the current United Nations agenda of 17 Sustainable Development Goals (SDGs) includes the goal of "ensuring healthy lives and promoting well-being for all at all ages", as well as reducing premature deaths due to non-communicable diseases by a third through prevention and treatment by 2030 [3]. Monitoring NCD morbidity and mortality is necessary to ascertain progress on national and global efforts. However, timely and accurate population-level health data remains limited in many LMICs due to lack of unbiased data collection systems [4].

Many LMICs lack civil registration and vital statistics systems and depend on mortality estimates from hospitals. However, hospital-based estimates do not give a complete picture on mortality as these facilities do not collect information on deaths occurring outside the healthcare system [5–7]. Given the challenges around civil registration and vital statistics in LMICs, health and demographic surveillance systems (HDSS) present an important source of information about the health of communities. A health and demographic surveillance system is a platform that collects longitudinal data, and monitors demographic transitions and health indicators among individuals within a well-defined geographic area [1].

A typical health and demographic surveillance system conducts verbal autopsies (VA) to generate cause-of-death data where standardized interviews are held with carers or close relatives of deceased persons to understand the circumstances surrounding the deaths [8,9]. The tool used to carry out these interviews is detailed and provides the interviewer an opportunity to get more in-depth information about the deceased. Questions are focused on the deceased's history, tracing from birth, to any illness that led to death, care-seeking patterns, challenges involved in getting treatment, and place of death. Interviews are then analysed to ascertain the most probable cause of death [10,11]. Leveraging on the capacity of HDSS to capture deaths, these systems present useful platforms for understanding mortality through longitudinal surveillance.

The present analysis is based on cause-of-death data generated through verbal autopsies conducted in Iganga-Mayuge Health and Demographic Surveillance Site (IMHDSS) in Eastern Uganda. We assessed mortality rates due to NCDs among persons ≥ 30 years old for the period 2010 to 2016. We were particularly interested in understanding mortality patterns due to cardiovascular diseases (CVD), cancers, and diabetes, conditions which are leading contributors to NCD mortality globally and within sub-Saharan Africa [1].

## Materials and methods

### Study area and population

The Iganga-Mayuge HDSS is located in the Iganga and Mayuge districts in Eastern Uganda and was established in 2005. The demographic surveillance area consists of 65 villages spread over a 155km$^2$ area with a population of 94,568 at the end of 2017. The average household size

is five individuals, and the area is predominantly rural, with some peri-urban areas. Subsistence agriculture is the main occupation and sex distribution is roughly equal, with 51% female. Approximately 40% of the population is less than 15 years old. The Iganga-Mayuge HDSS collects longitudinal data on births, deaths (and causes of deaths), and migrations; and monitors key interventions at a community-level [12].

## Verbal autopsy

Verbal autopsy is an epidemiological tool used to assess cause of death in settings where reliable civil registration and vital statistics systems are lacking [13,14]. Iganga-Mayuge HDSS conducts verbal autopsies to ascertain causes of death in the population. Data collectors report deaths that occur in the surveillance communities to the HDSS offices during routine data collection rounds conducted every six months. As of 2017, the overall response rate in the HDSS survey rounds was over 75%, after accounting for vacant homes (10.4%), demolished homes (5.8%), changed status of premises (5.5%), those not found at home (0.2%) and refusals (0.04%) [12]. The Verbal Autopsy Structured Questionnaire (developed by World Health Organisation in 2014 and standardized by the HDSS) is used to conduct face-to-face interviews with relatives or carers of the deceased [15]. The questionnaire requests information about the circumstances surrounding the death as well as healthcare sought.

**Classification of causes of death.** Two physicians independently review verbal autopsies and assign cause of death using the International Statistical Classification of Diseases and Related Health Problems (ICD-10). In case of disagreements, both physicians hold a consensus meeting to compare results and establish the most probable cause of death. In case of further disagreement on cause of death, a third physician breaks the tie [16,17]. For the present analysis, information on deaths among persons aged $\geq$ 30 years old was extracted from the Iganga-Mayuge HDSS database.

## Statistical analyses

Age-adjusted mortality rates (AAMR) were calculated using direct method, with the average population across the seven years of the study (2010 to 2016) as the standard. We used age categories of 30–40, 41–50, 51–60, 61–70, and $\geq$ 71 years for the standardization. For these calculations, the numerator was the number of deaths among persons $\geq$ 30 years old while the denominator was total number of people $\geq$ 30 years old in the population at midyear. Stata version 14 was used for analyses [18].

**Outcome definitions.** First, causes of death were broadly classified as communicable diseases, non-communicable diseases and injuries. Communicable diseases included those diseases that can be transmitted including: HIV/AIDS, pulmonary tuberculosis, malaria, acute febrile illnesses, diarrhoeal diseases, pneumonia and acute respiratory infections. Non-communicable diseases included those conditions that are non-infectious and typically of a chronic duration including: heart disease, diabetes, cardiovascular diseases, renal disorders, cancers, abdominal conditions and central nervous system disorders. Injuries consisted of accidental and intentional deaths including: accidental poisoning, animal bite/attack, drowning, falls, homicidal injuries, road traffic accidents, and suicidal injuries. As our focus is on major non-communicable diseases (NCDs), we further classified this category into cardiovascular diseases, cancers, diabetes, acute abdominal conditions, and other NCDs.

## Ethical considerations

Ethical approval was received from the Makerere University School of Public Health Research and Ethics Committee (MakSPH IRB 042) and the Uganda National Council of Science and

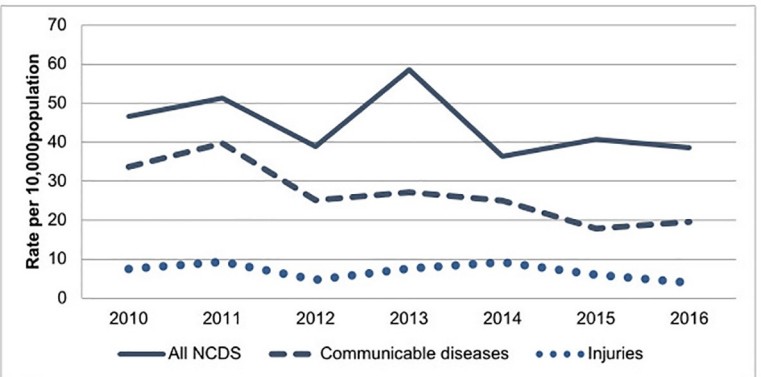

**Fig 1. Age-adjusted mortality rates of three major disease categories (communicable diseases, NCDs, and injuries) in IM-HDSS, expresses per 10,000 persons, 2010–2016.**

Technology (UNCST SS2002). Informed consent was received from relatives or carers of the deceased who were ≥ 18 years of age, and data was anonymized before analysis. All respondents were told about the content and purpose of the data collection.

## Results

### All NCD mortality

A total of 1,210 deaths among persons ≥ 30 years old were reported from 2010 to 2016. Of the total number of deaths under consideration, 50.7% were among women. The mean age of the deceased was 49.3 years, with a standard deviation of 14.4 years. Approximately 53% of all deaths among persons ≥ 30 years were due to non-communicable diseases, 31.8% due to communicable diseases, 8.2% due to injuries, and 7% due to maternal-related deaths or undetermined causes. Fig 1 shows the age-adjusted mortality rates from NCDs, communicable diseases and injuries for each of the seven years under consideration.

As shown in Table 1, cardiovascular diseases accounted for the largest proportion of reported NCD deaths in each year. There were substantial fluctuations in the rates of cardiovascular disease deaths between 2010 and 2016. Specifically, between 2010 and 2011, cardiovascular disease mortality rates (per 10,000 persons) increased by 71.2% (15.52 vs. 26.57), and

**Table 1. Age-adjusted mortality rates from major non-communicable diseases in Iganga-Mayuge health and demographic surveillance site, expressed per 10,000 persons, 2010–2016.**

| | All | | Cardiovascular diseases | | Diabetes | | Cancers | | Acute abdominal conditions | | Other NCDs | |
|---|---|---|---|---|---|---|---|---|---|---|---|---|
| Year | Population, number | Deaths | No. of deaths | AAMR | No. of deaths | AAMR | No. of deaths | AAMR | No. of deaths | AAMR | No. of deaths | AAMR |
| 2010 | 16,982 | 180 | 28 | 15.52 | 8 | 4.47 | 13 | 7.32 | 13 | 7.40 | 21 | 11.89 |
| 2011 | 17,850 | 208 | 50 | 26.57 | 14 | 7.49 | 9 | 4.80 | 7 | 3.84 | 16 | 8.61 |
| 2012 | 18,634 | 149 | 24 | 12.55 | 10 | 5.26 | 7 | 3.68 | 18 | 9.48 | 15 | 7.91 |
| 2013 | 19,650 | 196 | 60 | 30.33 | 18 | 9.08 | 15 | 7.58 | 16 | 8.06 | 7 | 3.53 |
| 2014 | 20,619 | 152 | 30 | 15.02 | 13 | 6.46 | 10 | 5.01 | 10 | 4.93 | 10 | 4.96 |
| 2015 | 21,375 | 165 | 31 | 14.91 | 22 | 10.59 | 9 | 4.21 | 12 | 5.81 | 11 | 5.25 |
| 2016 | 22,749 | 160 | 37 | 16.78 | 17 | 7.78 | 9 | 4.13 | 12 | 5.35 | 10 | 4.50 |

AAMR: Age-adjusted mortality rates; NCDs: Non-Communicable Diseases.

Age-adjusted mortality rates are expressed per 10,000 persons.

**Table 2. Age-adjusted mortality rates by gender, and expressed per 10,000 persons, 2010–2016.**

|  | 2010 | 2011 | 2012 | 2013 | 2014 | 2015 | 2016 |
|---|---|---|---|---|---|---|---|
| **Cardiovascular disease** |  |  |  |  |  |  |  |
| Total | 15.52 | 26.57 | 12.55 | 30.33 | 15.02 | 14.91 | 16.78 |
| Men | 9.49 | 21.11 | 6.54 | 15.73 | 10.15 | 7.90 | 11.28 |
| Women | 21.38 | 31.59 | 18.15 | 43.91 | 19.68 | 21.48 | 21.86 |
| **Diabetes** |  |  |  |  |  |  |  |
| Total | 4.47 | 7.49 | 5.26 | 9.08 | 6.46 | 10.59 | 7.78 |
| Men | 4.71 | 6.58 | 6.56 | 11.46 | 9.13 | 10.00 | 11.44 |
| Women | 4.22 | 8.30 | 4.06 | 6.86 | 3.91 | 11.14 | 4.39 |
| **Cancers** |  |  |  |  |  |  |  |
| Total | 7.32 | 4.80 | 3.68 | 7.58 | 5.01 | 4.21 | 4.13 |
| Men | 5.84 | 5.47 | 2.20 | 4.17 | 5.18 | 3.05 | 4.85 |
| Women | 8.71 | 4.17 | 5.08 | 10.75 | 4.90 | 5.30 | 3.48 |
| **Acute abdominal conditions** |  |  |  |  |  |  |  |
| Total | 7.40 | 3.84 | 9.48 | 8.06 | 4.93 | 5.81 | 5.35 |
| Men | 11.73 | 5.69 | 11.87 | 9.39 | 4.05 | 7.08 | 5.46 |
| Women | 3.33 | 2.06 | 7.19 | 6.81 | 5.76 | 4.65 | 5.26 |
| **Other NCDs** |  |  |  |  |  |  |  |
| Total | 11.89 | 8.61 | 7.91 | 3.53 | 4.96 | 5.25 | 4.50 |
| Men | 18.70 | 14.44 | 10.96 | 6.28 | 6.12 | 10.95 | 8.49 |
| Women | 5.52 | 3.13 | 5.07 | 0.98 | 3.83 | 0.00 | 0.80 |

decreased by over half between 2011 and 2012. Cardiovascular disease mortality rates also increased over two-fold between 2012 and 2013, from 12.55 to 30.33 (per 10,000 persons). Similar jumps were observed in diabetes mortality rates—between 2010 and 2011, the diabetes mortality rate increased 67.6% (4.47 vs. 7.49 per 10,000 persons). For the other years, mortality rates from cardiovascular disease remained relatively stable within the range of 12.55 to 16.78 deaths per 10,000 persons. No discernible trends were seen in overall mortality rates from cancers and acute abdominal conditions between 2010 and 2016. Age-adjusted mortality rates (per 10,000 persons) from other NCDs was highest in 2010 (11.89), then 2011 (8.61), and 2012 (7.91), and was in the range of 3.53 to 5.25 per 10,000 persons for the years 2013 to 2016.

**Gender differences in NCDs.** Table 2 shows gender and disease specific NCD mortality rates in 2010 and 2016. As shown in Table 2, women had higher mortality rates from cardiovascular diseases compared to men—in 2010 and 2016, mortality rates per 10,000 persons were 21.38 and 21.86 among women; and 9.49 and 11.28 among men (Fig 2). Across all the years, women had substantially higher cardiovascular disease mortality rates compared to men. Conversely, women had lower diabetes mortality rates than men for all years, except in 2011 and 2015. Women had higher mortality rates from cancers compared to men in 2010 (8.71 vs. 5.84 per 10,000 persons); however, cancer mortality rates were lower for both genders for the subsequent years (with the exception of the jump seen in women in 2013). Overall, men had higher rates of mortality from acute abdominal conditions compared to women. The rate of mortality from other NCDs was higher for men than for women in 2010, and lower for both genders in 2016 (Table 2).

## Discussion

We report on NCD mortality rates among persons ≥ 30 years obtained from verbal autopsy reports collected by the Iganga-Mayuge Health and Demographic Surveillance Site (IMHDSS)

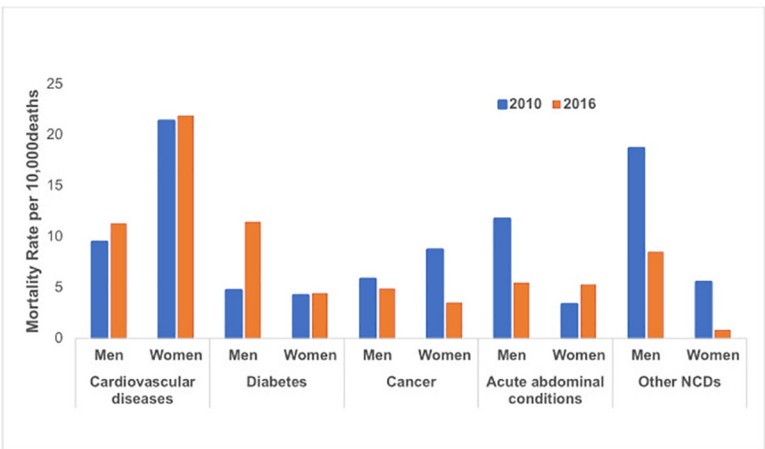

**Fig 2. Age-adjusted, gender-specific mortality rates in IM_HDSS (expressed per 10,000 persons), 2010 and 2016.**

in Eastern Uganda from 2010 to 2016. Cause of death was determined by verbal autopsies (i.e., physicians studied the history and circumstances surrounding each death to assign most probable cause of death based on ICD-10 codes), a method used to assign cause of death in the absence of well-established vital statistics systems [8,9]. In total, 1,210 deaths were reported between 2010 and 2016. Among broad categories of causes of death, NCDs were the leading cause, accounting for 53% of all deaths. Approximately 31.8% of all deaths were due to communicable diseases, 8.2% due to injuries, and 7% due to maternal-related deaths or of undetermined causes. Furthermore, cardiovascular diseases accounted for the largest proportion of NCD deaths, with substantially higher rates of cardiovascular disease deaths observed for women than for men.

Population-representative studies conducted within the Iganga-Mayuge health and demographic surveillance site show appreciable rates of major NCDs and NCD risk factors in the community. For example, in a study of men and women aged 35 to 60 years old conducted between March and April 2012, Mayega and colleagues observed a 7.4% prevalence of diabetes and 8.6% prevalence of pre-diabetes [19]. Moreover, 6.5% and 9.3% of men had diabetes and pre-diabetes respectively, while the percentages were 8.1% and 8.0% among women. In the same study, the authors noted a 20.5% prevalence of hypertension, 5.8% prevalence of current tobacco use, 4.8% prevalence of harmful use of alcohol, 12.6% prevalence of overweight, and 5.3% prevalence of obesity [19]. In another study examining overweight and obesity among persons aged $\geq$ 18 years and over, Kirunda and colleagues observed substantial prevalence of overweight and obesity, and significant gender differences. Using standard definitions of overweight (BMI between 25.0–29.99 kg/m$^2$), and obesity (BMI $\geq$ 30 kg/m$^2$), the authors estimated a 17.8% overweight prevalence (12.4% in men, 23.1% in women); and 7% obesity prevalence (2.0% in men, and 12.7% in women) [20]. Differences in the distribution of NCD risk factors across gender in this geographic area may play a role in observed differences in NCD mortality rates. Qualitative studies conducted within the Iganga-Mayuge area also show that myths and misconceptions about NCDs and their metabolic risk factors are commonplace [21]. Estimates of NCDs and NCD risk factors from studies conducted in Iganga-Mayuge HDSS are congruent with findings from the national non-communicable diseases risk factor survey published in 2015 which found a high prevalence of hypertension in the Eastern region of Uganda (26.4%) among persons 18 to 64 years old [22] Likewise, other studies conducted in rural and urban areas of Uganda showed substantial prevalence of NCDs and NCD risk factors [23–26].

In the national non-communicable diseases risk factor survey, prevalence of hypertension in the Northern, Central and Western regions of Uganda were 23.3%, 28.5%, and 26.3% respectively [22].

While peer-reviewed literature on rates of NCD morbidity and risk factors in Uganda have become more available over the last decade, there is much less information on NCD mortality rates in the country. However, emerging data indicate NCDs as major causes of hospitalization and in-hospital deaths. Investigators of a retrospective study published in 2019 examined medical records on admissions and mortality among in-patients at Mulago Hospital, Kampala, the largest public hospital in Uganda. They observed that an NCD was the primary reason for admission in 72% of patients between January 2011 and December 2014. Of 8,637 deaths that occurred during hospitalization, the following conditions had the highest case-fatality rates: non-tuberculosis pneumonia (28.8%), tuberculosis (27.1%), stroke (26.8%), cancer (26.1%) and HIV/AIDS (25%) [27].

Findings from neighbouring countries also show substantial prevalence of NCD risk factors and conditions, though much less information is available on NCD mortality. A study examining households in Nandi district, rural Kenya, and Dar es Salaam, urban Tanzania found age-standardized hypertension prevalence of 21.4% in the Kenyan sample and 23.7% on the Tanzanian sample [28]. Nationally representative data show a 15.4% prevalence of hypertension in Rwanda among persons aged 15–64 years [29]; and a prevalence of 24–25% among adults aged 25–64 years in Tanzania [30]. A study published in 2019 reported increasing disability-adjusted life years attributable to NCDs between 1990 and 2016 in Kenya [31].

While we are not able to validate the verbal autopsy reports within our study population, findings from some validation studies conducted within sub-Saharan Africa provide support for the utility of physician review of verbal autopsies for determination of NCD deaths. In a study of adult deaths at Kilifi District hospital in Kenya, the authors observed that hospital cause of death (HCOD) based on clinical and laboratory data (gold standard), a computer-based probabilistic model, and physician-certified verbal autopsy (PCVA) obtained the same underlying top five causes of deaths, and the kappa statistic for HCOD compared to PCVA was 0.52 (95% CI: 0.48–0.54). Furthermore, sensitivity for cardiovascular diseases was 70% and 100% for diabetes, while the specificities were both 96% [32]. In another study from Agincourt, South Africa, investigators observed sensitivity, specificity, and positive predictive values of 75%, 98% and 60% for NCDs among adults when comparing physician reviewed verbal autopsies to hospital records [33]. The Addis Ababa Mortality Surveillance Program observed sensitivity, specificity, and positive predictive values of 69%, 78% and 79% for NCDs when comparing physician reviewed verbal autopsies to hospital reports [34]. Physician-certified verbal autopsy has been more widely used for assessments of child mortality within sub-Saharan Africa, and investigators have noted that accuracy and reliability of verbal autopsies may be dependent on respondent factors (e.g., the level of healthcare they have access to), as well as physician knowledge about the epidemiology of disease in the geographic area [35]. These and other factors may influence accuracy and reliability of verbal autopsies in which NCDs are determined to be causes of death.

We cannot completely rule out or determine the extent of potential cause of death misclassification in our study, especially among individuals with multiple morbidities—this is a limitation. Relatedly, we only examined data for seven years within a relatively modest population, thus we were unable to establish clear patterns or trends due to lack of statistical power. Despite these limitations, our study provides much needed assessment of sub-national NCD mortality in Uganda, and at least two physicians reviewed each verbal autopsy report to adjudicate causes of deaths.

## Conclusions

From these analyses, it emerges that NCDs are leading causes of death in the Iganga and Mayuge communities in Eastern Uganda, with cardiovascular diseases and diabetes being two predominant causes. Furthermore, there appear to be differences in NCD mortality rates between men and women. In the absence of well-established vital statistics systems, verbal autopsies serve an important role in enabling monitoring of trends and mortality levels in low-resource settings; the utility of these platforms can be further enhanced through assessments of their validity and reliability to identify opportunities for improvements. Our findings contribute to understanding of leading causes of death to inform investments in health systems financing, support for burden of disease research, disease prevention strategies, as well as policy and health systems improvement initiatives in Uganda. Efforts are needed to tackle NCD risk factors and NCD care to reduce the burden and premature mortality due to these conditions.

## Acknowledgments

We thank the people of Iganga and Mayuge districts for supporting the activities of Makerere University Center for Health and Population Research (MUCHAP). Special thanks go to MUCHAP staff for their efforts in this study. Finally, we acknowledge the INDEPTH Network for providing technical support towards improving cause-of-death analysis and reporting.

## Author Contributions

**Conceptualization:** Davis Natukwatsa, Dan Kajungu.

**Data curation:** Davis Natukwatsa.

**Formal analysis:** Davis Natukwatsa, Adaeze C. Wosu.

**Methodology:** Davis Natukwatsa, Adaeze C. Wosu, Dan Kajungu.

**Validation:** Davis Natukwatsa, Musa Waibi.

**Visualization:** Davis Natukwatsa.

**Writing – original draft:** Davis Natukwatsa.

**Writing – review & editing:** Davis Natukwatsa, Adaeze C. Wosu, Donald Bruce Ndyomugyenyi, Musa Waibi, Dan Kajungu.

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
