## [Decision Letter · Decision Letter 0]

1 Dec 2020

PONE-D-20-31885

An analysis of non-communicable disease mortality among adults in Eastern Uganda, 2010-2016

PLOS ONE

Dear Dr. NATUKWATSA,

Thank you for submitting your manuscript to PLOS ONE. After careful consideration, we feel that it has merit but does not fully meet PLOS ONE’s publication criteria as it currently stands. Therefore, we invite you to submit a revised version of the manuscript that addresses the points raised during the review process.

Please revise the manuscript and address all peer review comments (especially the quality of written English).

We look forward to receiving your revised manuscript.

Kind regards,

Amir Radfar, MD,MPH,MSc,DHSc

Academic Editor

PLOS ONE

Journal Requirements:

"We wish to thank Swedish International Development Cooperation Agency for supporting MUCHAP/IMHDSS research activities."

"The author(s) received no specific funding for this work"

Reviewers' comments:

Reviewer's Responses to Questions

**Comments to the Author**

1. Is the manuscript technically sound, and do the data support the conclusions?

Reviewer #1: No

Reviewer #2: Yes

Reviewer #3: Yes

Reviewer #4: Yes

Reviewer #5: Yes

2. Has the statistical analysis been performed appropriately and rigorously? 

Reviewer #1: No

Reviewer #2: Yes

Reviewer #3: Yes

Reviewer #4: Yes

Reviewer #5: Yes

3. Have the authors made all data underlying the findings in their manuscript fully available?

Reviewer #1: Yes

Reviewer #2: Yes

Reviewer #3: Yes

Reviewer #4: Yes

Reviewer #5: Yes

4. Is the manuscript presented in an intelligible fashion and written in standard English?

Reviewer #1: No

Reviewer #2: Yes

Reviewer #3: Yes

Reviewer #4: Yes

Reviewer #5: Yes

5. Review Comments to the Author

Reviewer #1: 1. The authors have included adults above 30 years in the study. Adults above 60 years are considered Elderly population. Elderly population usually have many NCDs.

2. Abstract : Keywords : NCD has been repeated.

3. Introduction : Sex may be replaced by Gender in the text and Tables.

4. References - Although relevant references have been cited but they are not uniform.

5. Ref.Nos.7, Name of the journal is missing.

6. Some references are quite old. If possible, cite recent ones.

7. The authors have cited references from Uganda and Nairobi. Is NCD not a problem of other countries ? Try to include relevant references from studies in other countries.

Reviewer #2: Dear Authors,

Thank you for Very interesting and important work on NCD mortality among the among adults in Eastern Uganda.

Abstract, title and references

Yes; the aim of the study is clear. It is very clear what the study found and how they did it.; Yes; the title of the study is relevant, self-explanatory and informative but I would like to suggest to omit an analysis of from the title and the title might be "Non-communicable disease mortality among adults in Eastern Uganda, 2010-2016". References are relevant and recent. Authors used the references very correctly and they included all the appropriate key studies.

Introduction/background

Introduction is explained nicely and justify the importance of the study. It is also explaining the topic in different dimension.; Yes; research question is outlined clearly and aligned with the research problem.

Methods

Yes; the process of subject selection is clear’.; Yes; this study can be replicated easily with the existing methodology written by the authors.

Results

Yes; data is presented in an appropriate way and tables are relevant and clearly presented but it will be better if the trends can be presented through 2 to 3 graphs. Titles, columns, and rows labelled correctly and clearly.; Yes; categories grouped appropriately in the tables.; Yes; the text in the results add to the data and is not repetitive and critically discussed in to the text.; Yes; I am clear about what is a statistically significant result.; Yes; I am also clear enough about what is a practically meaningful result.

Discussion and Conclusions

Yes; the results discussed from multiple angles and placed into context without being overinterpreted.; Yes; the conclusions answered the aims of the study.; Yes; the conclusions supported by results.; No; the limitations of the study are not fatal; and Yes; they are opportunities to inform future research

Major Comments

-----------------

1. Yes, the study design appropriate to answer the aim

2. Yes, this study adds something new that is not known on this topic.

3. Yes, the article is consistent within itself

Minor Comments

-----------------

Need to trend analysis to understand the NCD mortality pattern. Now it is just distribution of NCDs deaths.

Reviewer #3: PLOS REVIEW REPORT

SUMMARY

Use of verbal autopsy to assess premature NCD mortality in resource-poor settings such as rural Eastern Uganda appears appropriate in the absence of country wide civil registration and vital statistics.This study adds to the growing premature NCD mortality data in Africa, even in rural areas.

MAJOR ISSUES

Lack of comparison of these results with the INDEPTH Network data from multiple sites in Africa.

Lack of comparison with epidemiology studies in rural and urban Uganda and and rest of the East Africa region

Wide unexplained disparity and fluctuations in mortality rates in different years especially in CVD and diabetes as well as between males and females. Raises questions about the reliability and accuracy of the data.

Lack of explanation with the unexpectedly high NCD mortality rates in a rural area in the presence of low NCD risk factors such as obesity rates (5.3%) and other risk factors.

In other studies in sub-Saharan Africa, some of the sites reporting the highest rates of NCD mortality were also those with the greatest burden of HIV/AIDS-related mortality. It has been suggested that in such settings, around half of the mortality attributed to NCDs may well be associated with HIV. It is not clear how the authors dealt with this.

MINOR ISSUES

1. The use of a probabilistic model to assign cause of death would have been preferable to

the use of doctors.

OTHER COMMENTS

Very useful source of information in the absence of inadequate data in the rest of the country.

Similar longitudinal studies would be useful in monitoring NCD mortality trends in other areas of the country to determine more reliable national trends.

Despite the HIV, Ebola and the COVID-19 epidemics, for the present as well as the foreseeable future, NCDs will be responsible for the highest morbidity and mortality in sub-Saharan Africa.The findings of studies like this, should spur policy makers to take action to reduce premature adult mortality due to NCDs.

Reviewer #4: Title: An analysis of non-communicable disease mortality among adults in Eastern Uganda, 2010-2016

1. Do the title and abstract accurately convey what has been found?

Yes, title is ok, but I would use the word “assessment” instead of analysis – this is usually reserved for a novel way of analysis e.g., new statistical analysis method

2. Is the writing acceptable?

Yes, writing is acceptable.

3. Are the data sound?

Yes, steps followed look sound and well documented

4. Do the figures appear to be genuine, i.e., without evidence of manipulation?

Yes, figures appear genuine

5. Does the manuscript adhere to the relevant standards for reporting and data deposition?

Yes, standards are adhered to

6. Are the discussion and conclusions well balanced and adequately supported by the data?

Yes, but discussion is quite light might need some beefing up.

7. Are limitations of the work clearly stated?

Yes, study limitations are clearly stated.

8. Do the authors clearly acknowledge any work upon which they are building, both published and unpublished?

Yes, this is done well in the discussion perhaps a little more in the introduction.

9. General Observation

Overall, it is a well described and interesting paper that adds to the current body of evidence and speaks to the current NCD burden especially NCD mortality within sub-Saharan Africa generally and Uganda specifically.

10. Minor Essential Revisions

• Classification of outcomes - indicate how multiple causes of death or multiple conditions present at time of death were handled and death classified in that case e.g., if at death someone had both Cancer and DM or had both DM and accidental death

• Need to reformat the tables to highlight points of interest e.g., bolding or re-aligning or smaller font size for some figures e.g. Table 2 can be better displayed as below

2016 2015 2014

Men Women Overall Men Women Overall Men Women Overall

11.29 19.68 16.78 7.9 21.48 14.91 10.15 21.86 15.02

• Indicate the AAMR units e.g. is it deaths per 10,000 or per 100,000?

• Built up the discussion nicely to indicate the burden and CV risk factor profile in Iganga-Mayuge - but do but not tie it up to show how this would lead to a high CVD mortality e.g.

o What are the known mortality trends in the region if any e.g. Eastern Uganda, in Uganda or E. Africa?

o How do these trends compare with what was found?

o How do they compare with regional hospital mortality trends, any surprises there??

o What might account for the anomalies e.g., fewer deaths in females, fewer deaths to DM in 2016 etc..

o How might these data on mortality be useful going forward, perhaps operationally??

11. Level of interest

I think this is a worthwhile study and a timely addition of valuable information to the current body of evidence with regard to NCD mortality in sub-Saharan Africa.

12. Quality of written English

Acceptable

13. Declaration of competing interests

I declare that I have no competing interests

Reviewer #5: The manuscript is technically sound as a verbal autopsy, a well-established method for such study, was utilized. The data also support the major conclusion of the manuscript that NCD is a major cause of death among the study population. However, the recommendation “…….a unified approach towards disease prevention and treatment that focuses on strengthening health systems” seems far fetching as the finding did not allude to any health system issues.

6. PLOS authors have the option to publish the peer review history of their article (what does this mean?). If published, this will include your full peer review and any attached files.

Reviewer #1: **Yes: **TAHZIBA HUSSAIN

Reviewer #2: **Yes: **Palash Chandra Banik

Reviewer #3: **Yes: **Prof. Peter Lamptey

Reviewer #4: No

Reviewer #5: **Yes: **DR. FEKADU ADUGNA DADI, PUBLIC HEALTH SPECIALIST

---

## [Author Response · Author response to Decision Letter 0]

18 Feb 2021

Review Comments to the Author

Reviewer #1: 

1. The authors have included adults above 30 years in the study. Adults above 60 years are considered Elderly population. Elderly population usually have many NCDs.

Thank you for this comment. Given our modest population, and observations that individuals in low- and middle-income country settings tend to develop and die from NCDs at an earlier age than individuals in high income country settings, we decided to examine NCD mortality in persons aged 30 years and above within our setting. Additionally, we performed age-adjustment to account for confounding by age.

2. Abstract: Keywords: NCD has been repeated.

We have deleted the repeated word.

3. Introduction: Sex may be replaced by Gender in the text and Tables.

The tables and text have been updated as suggested. 

4. References - Although relevant references have been cited but they are not uniform.

Thank you for this comment. We have adjusted the references so that they are uniform.

5. Ref.Nos.7, Name of the journal is missing.

Name of journal has now been added to Reference No. 7.

6. Some references are quite old. If possible, cite recent ones.

Thank you for this comment. We have included more recent citations where possible. 

For example: Kajungu D, Hirose A, Rutebemberwa E, Pariyo GW, et al. Cohort Profile: The Iganga-Mayuge Health and Demographic Surveillance Site, Uganda (IMHDSS, Uganda). International Journal of Epidemiology 2020 has been referenced in the materials and methods sections. Similarly, Nahimana MR, Nyandwi A, Muhimpundu MA, Olu O, et al. A population-based national estimate of the prevalence and risk factors associated with hypertension in Rwanda: implications for prevention and control. BMC Public Health 2018 has been cited under the discussion section.

7. The authors have cited references from Uganda and Nairobi. Is NCD not a problem of other countries? Try to include relevant references from studies in other countries.

Thank you for this comment. More references from sub-Saharan Africa reporting findings from neighboring countries (Kenya, Rwanda, and Tanzania) have now been added to the discussion section.

Reviewer #2: 

We thank the reviewer for recognizing the merits of our study. Below, we have provided point-by-point responses to the reviewer comments

Major Comments

-----------------

1. Yes, the study design appropriate to answer the aim

2. Yes, this study adds something new that is not known on this topic.

3. Yes, the article is consistent within itself

We thank the reviewer for recognizing the merits of our study

Minor Comments

-----------------

Need to trend analysis to understand the NCD mortality pattern. Now it is just distribution of NCDs deaths.

We agree that a trend analysis would be helpful. However, due to few years of data under consideration (only seven years’ worth of data), the modest size of our population, and the few deaths observed in each year, we decided to focus on NCDs assessment with a possibility of doing trend analyses in subsequent publications when there is enough data.

Reviewer #3: PLOS REVIEW REPORT

SUMMARY

Use of verbal autopsy to assess premature NCD mortality in resource-poor settings such as rural Eastern Uganda appears appropriate in the absence of country wide civil registration and vital statistics. This study adds to the growing premature NCD mortality data in Africa, even in rural areas.

We thank the reviewer for recognizing the merits of our study

MAJOR ISSUES

Lack of comparison of these results with the INDEPTH Network data from multiple sites in Africa. 

We have now described findings from available peer-reviewed data from verbal autopsy validations and population-based NCD surveys in Africa within the discussion section. 

Lack of comparison with epidemiology studies in rural and urban Uganda and rest of the East Africa region

We have now described findings from other epidemiological studies on the prevalence of NCD risk factors and morbidity within Uganda within the discussion section. Unfortunately, what is lacking are studies about NCD mortality, where much work still remains to be done. This is one of the gaps we aim to help bridge with the current study.

Wide unexplained disparity and fluctuations in mortality rates in different years especially in CVD and diabetes as well as between males and females. Raises questions about the reliability and accuracy of the data.

We agree with the reviewer that these disparities and fluctuations raise concerns. However, this is to be expected as our study was conducted in a very modest population over a relatively short period of time. Thus, even a few deaths can greatly influence the estimates in a given year. What we hope readers take away is that the findings show that NCDs contribute to a large proportion of deaths among adults, and there appear to be differences in some NCD mortality patterns between men and women. These differences warrant further exploration. In addition, interventions are needed to reduce NCD risk factors, morbidity, and mortality.

Lack of explanation with the unexpectedly high NCD mortality rates in a rural area in the presence of low NCD risk factors such as obesity rates (5.3%) and other risk factors.

Thank you for this comment. We note that other studies have shown substantial prevalence of various NCD risk factors in the Iganga and Mayuge communities. However, the mechanisms underlying the NCD mortality rates observed in our setting are not fully understood. We hope to study the potential social and metabolic mechanisms behind NCD mortality rates in this population in greater detail in future. 

In other studies in sub-Saharan Africa, some of the sites reporting the highest rates of NCD mortality were also those with the greatest burden of HIV/AIDS-related mortality. It has been suggested that in such settings, around half of the mortality attributed to NCDs may well be associated with HIV. It is not clear how the authors dealt with this.

Thank you for this comment. However, we were unable to comprehensively examine the influence of HIV; this is a limitation of our study. 

MINOR ISSUES

1. The use of a probabilistic model to assign cause of death would have been preferable to

the use of doctors.

We would like to thank the reviewer for this comment. The use of physician review to assign cause of death is the method used by the Iganga-Mayuge Health and Demographic Surveillance Site, and thus what our analysis was based upon. 

OTHER COMMENTS

Very useful source of information in the absence of inadequate data in the rest of the country.

Similar longitudinal studies would be useful in monitoring NCD mortality trends in other areas of the country to determine more reliable national trends.

Despite the HIV, Ebola and the COVID-19 epidemics, for the present as well as the foreseeable future, NCDs will be responsible for the highest morbidity and mortality in sub-Saharan Africa. The findings of studies like this, should spur policy makers to take action to reduce premature adult mortality due to NCDs.

We thank the reviewer for recognizing the merits of our study.

Reviewer #4: Title: An analysis of non-communicable disease mortality among adults in Eastern Uganda, 2010-2016

1. Do the title and abstract accurately convey what has been found?

Yes, title is ok, but I would use the word “assessment” instead of analysis – this is usually reserved for a novel way of analysis e.g., new statistical analysis method

Thank you for this comment. We have now modified the title and used the word assessment instead of analysis

2. Is the writing acceptable?

Yes, writing is acceptable.

3. Are the data sound?

Yes, steps followed look sound and well documented

4. Do the figures appear to be genuine, i.e., without evidence of manipulation?

Yes, figures appear genuine

5. Does the manuscript adhere to the relevant standards for reporting and data deposition?

Yes, standards are adhered to

6. Are the discussion and conclusions well balanced and adequately supported by the data?

Yes, but discussion is quite light might need some beefing up.

We have added additional information within the discussion.

7. Are limitations of the work clearly stated?

Yes, study limitations are clearly stated.

8. Do the authors clearly acknowledge any work upon which they are building, both published and unpublished?

Yes, this is done well in the discussion perhaps a little more in the introduction.

Thank you for this comment and for acknowledging the work that has been put together.

9. General Observation

Overall, it is a well described and interesting paper that adds to the current body of evidence and speaks to the current NCD burden especially NCD mortality within sub-Saharan Africa generally and Uganda specifically.

10. Minor Essential Revisions

• Classification of outcomes - indicate how multiple causes of death or multiple conditions present at time of death were handled and death classified in that case e.g., if at death someone had both Cancer and DM or had both DM and accidental death

This is a good point! In scenarios where there were multiple causes of deaths, we relied on information from physicians who would compare results and establish the most probable cause of death. We have noted the inability to account for multiple causes of death as a limitation in our discussion section.

• Need to reformat the tables to highlight points of interest e.g., bolding or re-aligning or smaller font size for some figures e.g. Table 2 can be better displayed as below

2016 2015 2014

Men Women Overall Men Women Overall Men Women Overall

11.29 19.68 16.78 7.9 21.48 14.91 10.15 21.86 15.02

We appreciate the advice. After trying a number of different reformats, we made the decision to leave Table 2 as it was, in order to allow the reader easily have information to compare across the years and across genders. The table is to enable the reader to reference/confirm the results we report in the text, e.g., cardiovascular disease death rates are higher for women than for men. 

• Indicate the AAMR units e.g. is it deaths per 10,000 or per 100,000?

We have indicated the units in the title for the table due to limited space within the table. We also included a note in the last row of the table to state that the age-adjusted mortality rates are expressed per 10,000 persons.

• Built up the discussion nicely to indicate the burden and CV risk factor profile in Iganga-Mayuge - but do but not tie it up to show how this would lead to a high CVD mortality e.g.

o What are the known mortality trends in the region if any e.g. Eastern Uganda, in Uganda or E. Africa?

Thank you for the comment and advice. We have made amendments to the discussion section. Unfortunately, studies are still limited about NCD mortality in Uganda and neighboring countries, especially within population-based settings. This is one of the gaps we aim to help bridge with the current study. 

o How do these trends compare with what was found?

The trends are congruent with findings of a substantial prevalence of NCD risk factors and disease burden in the region. Although there is a dearth of hospital or community-based studies on NCD mortality in the East Africa region, we identified a study reporting on disability adjusted life years (DALYs) due to NCDs in Kenya for the period of 1990-2016; this study found increasing DALYs over the time period. We have referenced the study within the discussion section. In addition, we have referenced the NCD mortality trends study we identified which was based on four-year retrospective data from Mulago Hospital.

o How do they compare with regional hospital mortality trends, any surprises there??

Please see comment above

o What might account for the anomalies e.g., fewer deaths in females, fewer deaths to DM in 2016 etc..

Fluctuations and anomalies may be partly explained by our small sample size. Thus, after stratification by year, even a few deaths can have great influence on estimated mortality rates. We are careful about our speculations about the observed anomalies, as they require further study to provide explanations. 

o How might these data on mortality be useful going forward, perhaps operationally??

Thank you for this comment. The findings provide a useful baseline from which to track future trends in NCD mortality and to add knowledge to the existing literature on NCD mortality in Uganda and sub-Saharan Africa more generally, where there is currently limited data on NCD mortality.

11. Level of interest

I think this is a worthwhile study and a timely addition of valuable information to the current body of evidence with regard to NCD mortality in sub-Saharan Africa.

12. Quality of written English

Acceptable

13. Declaration of competing interests

I declare that I have no competing interests

Reviewer #5: The manuscript is technically sound as a verbal autopsy, a well-established method for such study, was utilized. The data also support the major conclusion of the manuscript that NCD is a major cause of death among the study population. However, the recommendation “…….a unified approach towards disease prevention and treatment that focuses on strengthening health systems” seems far fetching as the finding did not allude to any health system issues.

Thank you for this comment. The statement has been revised accordingly.

---

## [Decision Letter · Decision Letter 1]

9 Mar 2021

An assessment of non-communicable disease mortality among adults in Eastern Uganda, 2010-2016

PONE-D-20-31885R1

Dear Dr. NATUKWATSA,

We’re pleased to inform you that your manuscript has been judged scientifically suitable for publication and will be formally accepted for publication once it meets all outstanding technical requirements.

Kind regards,

Amir Radfar, MD,MPH,MSc,DHSc

Academic Editor

PLOS ONE

Additional Editor Comments (optional):

Reviewers' comments:

Reviewer's Responses to Questions

**Comments to the Author**

1. If the authors have adequately addressed your comments raised in a previous round of review and you feel that this manuscript is now acceptable for publication, you may indicate that here to bypass the “Comments to the Author” section, enter your conflict of interest statement in the “Confidential to Editor” section, and submit your "Accept" recommendation.

Reviewer #1: All comments have been addressed

Reviewer #2: All comments have been addressed

Reviewer #4: All comments have been addressed

Reviewer #5: All comments have been addressed

2. Is the manuscript technically sound, and do the data support the conclusions?

Reviewer #1: Yes

Reviewer #2: Yes

Reviewer #4: Yes

Reviewer #5: Yes

3. Has the statistical analysis been performed appropriately and rigorously? 

Reviewer #1: Yes

Reviewer #2: Yes

Reviewer #4: Yes

Reviewer #5: Yes

4. Have the authors made all data underlying the findings in their manuscript fully available?

Reviewer #1: Yes

Reviewer #2: Yes

Reviewer #4: Yes

Reviewer #5: Yes

5. Is the manuscript presented in an intelligible fashion and written in standard English?

Reviewer #1: Yes

Reviewer #2: Yes

Reviewer #4: Yes

Reviewer #5: Yes

6. Review Comments to the Author

Reviewer #1: I have gone through the revised manuscript entitled, “An analysis of non-communicable disease mortality among adults in Eastern Uganda,2010-2016” carefully.

The authors have incorporated the changes as suggested in the last review.

With all good wishes,

Reviewer #2: Now the manuscript is more improved and can be published in the journal in the present form. It is a very interesting and important study from Uganda. We know that we have very limited data from African region particularly in younger age group people. Best wishes.

Reviewer #4: I am satisfied with the changes made and responses to my original review. With the exception of the table alignments (which they have justified somewhat). I think the authors have done good job at addressing the issues I raised.

Reviewer #5: The manuscript is presented well and written in standard English. However, it needs proofreading (edits) as there are some grammatical errors. Eg. on page 12, fourth line from bottom, the phrase "........aged ≥ 18 years and over......." is redundant (the symbol is not required).

7. PLOS authors have the option to publish the peer review history of their article (what does this mean?). If published, this will include your full peer review and any attached files.

Reviewer #1: **Yes: **TAHZIBA HUSSAIN

Reviewer #2: **Yes: **Palash Chandra Banik

Reviewer #4: No

Reviewer #5: No

---

## [Editor Report · Acceptance letter]

12 Mar 2021

PONE-D-20-31885R1 

An assessment of non-communicable disease mortality among adults in Eastern Uganda, 2010-2016 

Dear Dr. Natukwatsa:

I'm pleased to inform you that your manuscript has been deemed suitable for publication in PLOS ONE. Congratulations! Your manuscript is now with our production department. 

Kind regards, 

on behalf of

Dr. Amir Radfar 

Academic Editor

PLOS ONE